# Multiscale Femoral Neck Imaging and Multimodal Trabeculae Quality Characterization in an Osteoporotic Bone Sample

**DOI:** 10.3390/ma15228048

**Published:** 2022-11-14

**Authors:** Enrico Soldati, Flavy Roseren, Daphne Guenoun, Lucia Mancini, Emilio Catelli, Silvia Prati, Giorgia Sciutto, Jerome Vicente, Stefano Iotti, David Bendahan, Emil Malucelli, Martine Pithioux

**Affiliations:** 1Aix Marseille University, CNRS, IUSTI, 13453 Marseille, France; 2Aix Marseille University, CNRS, CRMBM, 13385 Marseille, France; 3Aix Marseille University, CNRS, ISM, 13288 Marseille, France; 4Aix Marseille University, APHM, CNRS, ISM, Sainte Marguerite Hospital, Institute for Locomotion, Department of Radiology, 13274 Marseille, France; 5Elettra-Sincrotrone Trieste S.C.p.A, SS 14–km 1535 in Area Science Park, Basovizza, 34149 Trieste, Italy; 6Slovenian National Building and Civil Engineering Institute, Dimičeva ulica 12, 1000 Ljubljana, Slovenia; 7University of Bologna, Department of Chemistry “G. Ciamician”, Ravenna Campus, Via Guaccimanni, 42, 48121 Ravenna, Italy; 8Università di Bologna, Department of Pharmacy and Biotechnology (FaBit), Via Zamboni 33, 40126 Bologna, Italy; 9National Institute of Biostructures and Biosystems, Viale delle Medaglie d’Oro 305, 00136 Roma, Italy; 10Aix Marseille University, APHM, CNRS, ISM, Sainte-Marguerite Hospital, Institute for Locomotion, Department of Orthopaedics and Traumatology, 13274 Marseille, France

**Keywords:** Fourier transform infrared spectroscopy (FTIR), microindentation, magnetic resonance imaging (MRI), osteocytes lacunae, osteoporosis, X-ray computed microtomography (µCT)

## Abstract

Although multiple structural, mechanical, and molecular factors are definitely involved in osteoporosis, the assessment of subregional bone mineral density remains the most commonly used diagnostic index. In this study, we characterized bone quality in the femoral neck of one osteoporotic patients as compared to an age-matched control subject, and so used a multiscale and multimodal approach including X-ray computed microtomography at different spatial resolutions (pixel size: 51.0, 4.95 and 0.9 µm), microindentation and Fourier transform infrared spectroscopy. Our results showed abnormalities in the osteocytes lacunae volume (358.08 ± 165.00 for the osteoporotic sample vs. 287.10 ± 160.00 for the control), whereas a statistical difference was found neither for shape nor for density. The osteoporotic femoral head and great trochanter reported reduced elastic modulus (Es) and hardness (H) compared to the control reference (−48% (*p* < 0.0001) and −34% (*p* < 0.0001), respectively for Es and H in the femoral head and −29% (*p* < 0.01) and −22% (*p* < 0.05), respectively for Es and H in the great trochanter), whereas the corresponding values in the femoral neck were in the same range. The spectral analysis could distinguish neither subregional differences in the osteoporotic sample nor between the osteoporotic and healthy samples. Although, infrared spectroscopic measurements were comparable among subregions, and so regardless of the bone osteoporotic status, the trabecular mechanical properties were comparable only in the femoral neck. These results illustrate that bone remodeling in osteoporosis is a non-uniform process with different rates in different bone anatomical regions, hence showing the interest of a clear analysis of the bone microarchitecture in the case of patients’ osteoporotic evaluation.

## 1. Introduction

Bone is a dynamic tissue in which aged bone is being continuously resorbed and replaced by younger bone, so that mineral homeostasis and integrity of bone structure can be maintained [1]. This process, referred to as bone remodeling, is a multiscale process and recent works have highlighted the major role of osteocytes in maintaining bone mass and volume. Osteocytes, the most abundant bone cells, are stocked in lacunar voids distributed throughout the entire bone are interconnected by canaliculi and play a role in the anabolic response to mechanical stimuli [1,2]. This osteocytes-mediated mechanism, called mechano-transduction, leads to the release of soluble factors involved in bone resorption or bone formation [3,4]. Considering that osteocytes are considered to be key regulators of skeletal homeostasis, their role in bone diseases has been suggested through changes in morphology and network [5]. In that context, several imaging modalities have been developed for osteocytes. Conventional microscopic 2D imaging techniques are limited by the lack of the third dimension [6,7,8,9]. Multiple 3D imaging modalities have been used [10,11,12] and among them, synchrotron radiation computed tomography (SR CT) at the micro- and nanoscale levels is of interest, given that it can provide volumetric information and investigate the 3D material properties with a non-destructive approach [13,14,15,16]. Given the increasing interest in osteocytes, a large variety of imaging techniques providing different spatial resolutions have been used. However, data in human bone remain scarce and are generally limited to a very small field of view. On that basis, a complete morphological analysis of osteocytes lacunae (OL) would be of great interest, since changes could affect the bone remodeling process. Changes in lacunar shapes have been observed in pathologies such as osteopenia, osteopetrosis and osteoarthritis [17].

In particular, osteoporosis, one of the most common pathologies, is a systemic disease characterized by a reduced bone mineral density (BMD) and a thinning of both cortical and trabecular phase due to bone resorption, which usually leads to fragility fractures [17,18]. The increased susceptibility to fragility fractures usually accounts for the reduced bone strength and quality. While both determinants are tightly linked to several parameters related to bone geometry, architecture, mineralization, and remodeling, osteoporosis is still currently diagnosed on the basis of BMD measurements using dual-energy X-ray absorptiometry (DXA) [19]. BMD is considered to be an index of bone mass loss, and on that basis, as a predictor of the sensitivity to osteoporotic fractures. However, it has been acknowledged that bone strength depends not only on bone quantity, but also on its shape and hierarchical structure [19,20,21]. However, very few studies have assessed osteocytes morphology in osteoporosis [16].

Previous studies have shown that the bone post-yield behavior, characterizing the capacity to absorb the energy before failure, is linked to the organic phase and its alteration has been associated with changes in the mechanical behavior, making bones more ductile or brittle [22]. Moreover, the bone extracellular matrix (ECM), which plays an important role in the tissue rigidity, is highly related to the mechanical properties of the bone tissue [23]. In altered ECM, the mineral part of the bone will also be affected, leading to modulations in the mechanical behavior [22].

Recent studies have shown that the ECM is responsible for the mineralization process and influences the density of the crystal composition [24]. Hence, the bone multimodal analysis is of great interest since it would allow for characterization of the bone mechanical properties, level of mineralization, carbonate accumulation and collagen cross-links, providing reliable indicators of the bone material properties. The bone quality and mineralization have been also assessed using reference point indentation (RPI) [25], which is a microindentation technique able to directly measure mechanical properties in diseased bone samples [26,27,28,29]. However, further validation of this technique is required to assess the relation between RPI and bone material properties.

Although previous studies have assessed bone microarchitecture [21,30,31], composition [32,33,34,35], mechanical properties [36,37,38] and OL morphology and network [13,39,40], these characteristics have been usually investigated alone and in more accessible bone segments (cortical phase [37,41] tibiae [17,38], distal femur [31,32], vertebrae [34,36]). Therefore, an analysis combining all the previously presented characteristics on the trabeculae of same samples is of great interest to obtain a more comprehensive overview of the bone health state.

The present study aimed to characterize bone quality in the femoral neck region using a multiscale and multimodal approach including imaging by X-ray computed microtomography at different spatial resolutions, i.e., with a pixel size of 51.0, 4.95 and 0.9 µm, micro-indentation and Fourier transform infrared spectroscopy. The bone characterization of the osteocytes network, trabecular composition and mechanical properties could provide a better understanding of the bone quality, with direct implications in pre-clinical research, therapeutic strategy, and eventually clinical practice. We chose to assess the femoral neck region, given that femoral neck fractures amount to 14% of the overall occurring fragility fractures [18].

## 2. Materials and Methods

Femoral neck morphology was assessed using laboratory-based X-ray microtomography (µCT) and synchrotron radiation microtomography (SRµCT), providing an isotropic voxel size resolution of 51 and 4.95 µm, respectively. We also performed SRµCT scans of the femoral neck at a pixel size of 0.9 × 0.9 µm^2^ in order to obtain a 3D quantification of OL. The combination of µCT and SRµCT allowed the bone analysis at different scales, therefore focusing on volume of interests (VOI), in particular VOI equal to 66,325 mm^3^, 1042 mm^3^ and 6.27 mm^3^, respectively for isotropic voxel size resolutions of 51.0 µm, 4.95 and 0.9 µm. Low-resolution images were used for double purpose. First, image stacks have been visually inspected so as to identify all the target regions to be acquired at higher resolutions. These images were also analyzed so as to obtain all the derivable morphological information (see Appendix A). On that basis and considering that proximal femur fractures usually occur in the femoral neck, the analysis at 51.0 and 4.95 µm resolution was performed in the femoral neck. Then, the subsequent acquisitions at 0.9 µm resolution were performed in regions with the higher bone density according to the 4.95 µm resolution images. Finally, bone trabecular quality has been assessed using the microindentation and the Fourier transform infrared spectroscopy. The workflow of the current experimental study is presented in Figure 1. These methods provided information related to trabeculae mechanical properties, mineralization and carbonate accumulation in several anatomical regions of the osteoporotic proximal femur. The corresponding results were compared to those obtained in a healthy bone sample.

### 2.1. Sample Collection and Preparation

Bone collection was performed with the agreement of the local ethics committee and according to the 1975 Helsinki Declaration (revised in 2000). Both donors were female and, according to the DXA measurements, one bone sample was classified as osteoporotic, whereas the other one was classified as normal (Table 1). Samples were cut using a bandsaw along the axial direction right below the lesser trochanter (approximately 10 to 12 cm section proximal to the femur head), and the specimen were then stored at −25 °C.

### 2.2. Sample Extraction and Preparation for Microindentation Test

Using a drill press, equipped with a hollow tip (10 mm internal diameter) and under constant water irrigation, three small samples were extracted from each proximal femur so as to characterize the mechanical properties of the lamellar bone. These regions were chosen in the great trochanter (GT), the femoral neck (FN) and the femoral head (FH). The samples were harvested perpendicular to the coronal plane and kept frozen at −25 °C. The micro-indentation protocol was performed as previously described [42]. Briefly, the surface of each specimen was consecutively polished (ESC-200-GTL, ESCIL^®^ Chassieu, France) with carbide papers (P600, P1200, P2500) and multiple diamond slurries (6, 3, 1 and 0.25 µm). Prior to the mechanical assays, the sample preparation quality was validated using optical microscope Figure 2. Specimens were then stored in a solution containing calcium (50 mg/L) and sodium azide (0.01%) in order to prevent bone mineral matrix dissolution and collagen degradation [43]. They were kept refrigerated before indentation.

### 2.3. X-ray Microtomography Measurements

The μCT scans of both proximal femurs were acquired using a Rx-Solution EasyTom XL ULTRA instrument [44] as previously described [45].

Propagation-based phase contrast SRμCT data were used to obtain the 3D virtual reconstruction of the bone microstructure at the microscale. The central core across the whole length of femoral neck was imaged at the SYRMEP (SYnchrotron Radiation for MEdical Physics) beamline of the Elettra synchrotron facility (Basovizza [Trieste], Italy) using a filtered (1.5 mm Si +1 mm Al) polychromatic X-ray beam, delivered by a bending magnet source in transmission geometry and with a mean energy of 27 KeV. The detector used was a water-cooled, 16-bit, sCMOS macroscope camera (Hamamatsu C11440–22C) with a 2048 × 2048 pixels chip coupled with different GGG:Eu scintillator screens of different thickness, depending on the selected pixel size, through high numerical aperture optics. The effective pixel size of the detector was set at 4.95 × 4.95 µm^2^ and 0.9 × 0.9 μm^2^ using, respectively, a 45 and 17 µm thick scintillator screen, yielding a maximum field of view of about 10.1 × 10.1 mm^2^ and 1.8 × 1.8 mm^2^, respectively. The sample-to-detector (propagation) distance was set at 150 mm. A set of 1200 projections were recorded, with a continuous sample rotation over a 180-degree scan angle and an exposure time per projection of 2.5 s.

Each set of acquired raw images (projections) was processed using the SYRMEP Tomo Project (STP) software suite, developed in-house at Elettra [46] and based on the ASTRA Toolbox [47]. The tomographic reconstruction was performed using the filtered backpropagation algorithm, coupled to a filter [48] to reduce the so-called ring artefacts in the reconstructed slices. A single-distance phase retrieval algorithm was applied to projection images prior to reconstruction [49], setting the δ β parameter (ratio between the real and imaginary parts of the complex refraction index of the material under investigation) to 50 and 20 for images reconstructed with an isotropic voxel size of 4.95 and 0.9 µm^3^, respectively.

#### 2.3.1. Image Post-Processing and Analysis

The image post-processing and analysis have been performed using the software iMorph (iMorph_v2.0.0, AixMarseille University, Marseille, France) [50,51]. iMorph provides a 3D transformation map from which the aperture map can be derived. The aperture map provides, for each pixel of the bone/OL, the diameter of the maximal disk totally enclosed in the bone/OL and containing this voxel.

#### 2.3.2. SRµCT (Voxel Size: 0.9 µm)

Thanks to the extremely small voxel size (0.9 µm), the image field of view was focused on a single trabecula (Figure 3d) and the OL became assessable. First, a 3D median filter, using a window size of 3 × 3 × 3 voxels, has been applied to denoise the image volumes. In order to binarize the image stack, the threshold that better differentiated the bone from the pores (OL and vascular canals) was selected manually from the gray levels’ distribution of each volumetric image stack. Finally, a morphometric threshold of 27 voxels has been chosen to define the minimum number of voxels identifying an object, and these misclassified voxels were re-added to the solid phase. The bone porosity was measured as the percentage of the volume of the soft phase (small pores, OL and vascular canals) over the sum of the soft and bone phases. Second, the OL were identified as those regions composed by a volume in the 73 to 1000 µm^3^ range, chosen in accordance with the mean and standard deviation of OL previously found in the literature [4,14]. In addition, this strategy allowed us to distinguish the OL from the vascular canals network, which was characterized by higher volumes (Figure 3e,f) [14,15]. The image processing effect (median filtering, manual thresholding, and morphometric analysis) for the OL characterization has been investigated and is discussed in the Appendix B. Finally, each OL was individually characterized using the volume, surface, size, shape, sphericity, and fractal anisotropy and as an interconnected network by evaluating its region of action. In particular, in each OL the principal axes (a, b and c) of the maximum inscribed ellipsoid have been calculated. On that basis, the OL shape was computed from the derivation of the aspect ratios ($\frac{a}{b},\frac{b}{c},\frac{a}{c}\;\mathrm{and}\;\frac{a}{\left(b+c\right)}$). Then, the outer surface (S_OL_) and the sphericity index (Fsph) [52] were derived, respectively, by the formula:SOL=4∗π∗(apbp+apcp+bpcp)1p, with p=ln(3)/ln(2)
Fsph=6∗VOL∗(πSOL3)0.5
where $V_{OL}$ is the OL volume. Moreover, the fractal anisotropy (FrAn) [53], which reflects the axonal diameter and is an extension of the concept of eccentricity of conic sections in 3 dimensions, normalized to the unit range, has been calculated using the formula:FrAn=32∗(a−λ)2+(b−λ)2+(c−λ)2a2+b2+c2, whereλ=(a+b+c)3

Finally, the region of action could be considered as a 3D bone, neighboring the OL, in which the osteocytes sense and transmit information. It was derived by applying the Voronoi map, which calculates the minimum path between the closest OL and differs from the OL density because it is calculated for each OL singularly.

### 2.4. Microindentation

The trabecular bone of the whole set of specimens has been characterized using a microindentation apparatus (Tester NHT^2^, Anton Paar^®^, Switzerland and Austria) equipped with a sharp Berkovich diamond indenter (tip diameter: 120 nm, elastic modulus: 1141 GPa and Poisson’s ratio: 0.07) in a thermally controlled room at 23 °C and on a pneumatic antivibration table. For the indentation tests, the sample was set in a watertight support filled with calcium buffered saline up to the level of the polished surface. A total of 40 points were selected using a ×20 microscopic objective on five different trabeculae located on the polished surface of the sample. Points were placed in the trabecular centerline to avoid any border effect and verified at magnitude x100 to avoid any surface irregularities (i.e., porosity). Before each set of indentation, a calibration session test was made on a fused silica reference sample. The mean corresponding values were 71.3 ± 1.6 GPa while the reference data was 72 GPa.

A trapezoidal loading profile (30 s: 60 s: 30 s, max load 40 mN) has been applied, as shown in Figure 4, in the polished surface of each sample. The 60 s plateau time was chosen from preliminary tests because wet tissues display a more viscous mechanical behavior. The sample hardness (H) and elastic modulus (Es) were calculated using the Oliver and Pharr method [54]. In this study, the Poisson’s ratio was assumed to be 0.3, since the relative error, determined by varying the Poisson’s ratio from 0.2 to 0.4, was found to be less than 8% in elastic modulus [23,55].

The mean and standard deviation for elastic moduli and the hardness, both expressed in GPa, were obtained for the osteoporotic subject and the control.

### 2.5. Fourier Transform Infrared Spectroscopy

The attenuated total reflection Fourier transform infrared (ATR-FT-IR) measurements were obtained using a Thermo Nicolet iS50 FTIR spectrometer (Thermo Fisher Scientific Co., Waltham, MA, USA) equipped with a deuterated triglycine sulphate (DTGS) detector. The ATR spectra were recorded using a diamond ATR Smart OrbitTM accessory (from Thermo Optec) in the Mid-IR spectral range 4000–525 cm^−1^, averaging 64 scans for each measure and 64 scans for the background. The spectral resolution is 4 cm^−1^. For each bone sample, a small fragment was collected and analyzed. The fragments were first ground in an agate mortar, and the powder obtained was successively placed on the ATR diamond crystal. Pressure was applied on the powder to optimize the contact with the diamond crystal. Several replicate spectra were acquired for each fragment. The spectra selected for interpretation were those which presented the highest signal-to-noise ratio.

A qualitative analysis of the spectra has been conducted considering the range 527–1800 and 2550–4000 cm^−1^, where the most important features connected to bones component are visible. Generally speaking, the phosphate features of hydroxyapatite (Ca_10_(PO_4_)_6_(OH)_2_), the constituent of bones, show the main bands in the region 1050–1100 cm^−1^ (antisymmetric PO_4_^3−^ stretching $\nu_{a}$). Additional bands of the compound are at about 962 cm^−1^ (symmetric PO_4_^3−^ stretching, $\nu_{s}$) and between 660–520 cm^−1^ (in-plane PO_4_^3−^ bending mode, δ_ip_) [56,57,58]. Bones also have a carbonate group (CO_3_^2−^) allocated in the hydroxyapatite lattice. The infrared bands of carbonate are visible in the region 890–860 cm^−1^ (out-of-plane CO_3_^2−^ bending mode, $\delta_{oop}$) and 1400–1550 cm^−1^ (antisymmetric CO_3_^2−^ stretching, $\nu_{a}$) [57,59]. Apatite from natural bones may also contain a relevant amount of proteins. The presence of such an organic compound is visible from the CH and NH bands, respectively, at around 2800–3000 cm^−1^ and 3300 cm^−1^ (antisymmetric stretching, $\nu_{a}$) and additionally, from the more specific bands connected to the proteinaceous-based vibrational modes at 1637 (Amide I), 1550 (Amide II), and 1239 cm^−1^ (Amide III) [60,61,62].

It is well known that the carbonate group (CO_3_^2−^) in bones’ hydroxyapatite is present as a replacement of either PO_4_^3−^ or OH- groups. When the substitutions happen at PO_4_^3−^ sites, the compound is defined as B-type carbonated apatite (B-CAp); if the substitution is at OH- sites, the compound is an A-type carbonated apatite (A-CAp) [63,64]. According to several IR spectroscopic studies on synthetic apatite, the B-type is recognized by a doublet at ~1410 and ~1465 cm^−1^ (antisymmertric CO_3_^2−^ stretching, $\nu_{a}$) together with a signal at 870 cm^−1^ (out-of-plane CO_3_^2−^ bending mode, $\delta_{oop}$); conversely, the A-type is identified by a doublet at ~1456 and ~1540 cm^−1^ (antisymmertric CO_3_^2−^ stretching, $\nu_{a}$) and a signal at 880 cm^−1^ (out-of-plane CO_3_^2−^ bending mode, $\delta_{oop}$) [56,63,64,65]. The identification of A-type or B-type in biological apatite is more complex, as the IR characteristic signatures may be covered by the bands of other compounds. Ren at al. correctly pointed out that the peak at ~1546 cm^−1^, the characteristic signature of A-type, can be masked by the Amide II of collagen [56]. Moreover, the bands at 1410, 1455 and 880 cm^−1^ should be used carefully for the identification of Cap, since similar bands were observed in other carbonate species adsorbed onto the apatite surface and not in the lattice [56].

Band assignation is also of a certain difficulty for the A or B type when A and B carbonate are present together in the apatite lattice. Considering the 1400–1550 cm^−1^ region, a synthetic sample of the AB-type showed a single band at 1460 cm^−1^ that accounted for the contribution of two overlapped signals: the low-frequency $\nu_{a}$ band of the A doublet and the high-frequency $\nu_{a}$ band of the B doublet; the other $\nu_{a}$ bands revealed a shift in the wavenumbers compared to the pure B and A standards [64]. Similar assignation was considered by Pedrosa et al. for the bands of male and female femoral bone hydroxyapatite: the band at 1450 cm^−1^ is assigned to the type A + B carbonate, while the B type is recognized by the 1415 cm^−1^ band [66]. In some spectra of biological apatite, the above-mentioned band at 1460 cm^−1^ is observed as two signals: a band at 1450 and a shoulder at 1465 cm^−1^ [56]. According to several researchers, the signal at 1465 cm^−1^ may be assigned to the B-type and the one at 1450 cm^−1^ to the A type [56,58,67]. This assignation, however, is not without controversy [68,69]. Regarding the carbonate bending region between 860–890 cm^−1^ ($\delta_{oop}$ CO_3_^2−^), the AB-type usually shows a doublet; several researchers agreed to assign the band at 880 cm^−1^ to the A-type and the one at 872 cm^−1^ to the B-type [58,64,70]. The predominance of B-type in AB-type is possible, and was observed in a study conducted on cadaveric femur of a 39 yrs male [69]. In the study, Figueiredo et al. assigned the bands 1410, 1445 cm^−1^ and 871 cm^−1^ to the B-type apatite, while the band at 880 cm^−1^, shoulder, was assigned to type A [69].

### 2.6. Statistical Analysis

For the morphological parameters of each OL, student *t* tests were used to assess whether the osteoporotic samples differed significantly from the control samples. For each VOI, the potential relationships between the OL characteristics and the bone volume were assessed using the spearman coefficient of correlation (R^2^).

Regarding the trabeculae mechanical parameters, given the non-normal distribution identified through a Shapiro–Wilk test, non-parametric tests were used to assess the effects of ROIs (femoral head vs. femoral neck vs. great trochanter). Kruskal–Wallis tests and multiple pairwise comparisons with a Bonferroni correction were performed using Dunn’s tests. Mann–Whitney U tests were used to assess the differences between osteoporotic and control samples.

## 3. Results

### 3.1. Osteocytes Lacunae Characteristics

OL morphological parameters, computed from the OL embedded in the solid bone phase, are summarized in Table 2 and Figure 5. The number of OL per bone volume was slightly lower (−3%) in the osteoporotic subject as compared to the control, whereas the mean OL density showed the opposite effect, with the osteoporotic subject having a higher density (+3%) as compared to the control. The Voronoi map showed that the osteoporotic subject did not have a statistically significant lower region of action (−5%) than the control (56.7 × 103 µm^3^ vs. 59.6 × 103 µm^3^). Moreover, the mean OL volume of the osteoporotic subject (358.1 µm^3^) was statistically larger (25%) than the corresponding control value (287.1 µm^3^). The larger OL volume was also illustrated in the principal axes (a, b and c) of the maximum inscribed ellipsoid inside the OL, which resulted in non-statistically significant 8% larger axes in the osteoporotic OL as compared to the control. In addition, the osteoporotic OL surface was statistically larger compared to the controls (225.5 ± 13.7 µm^2^ vs. 195.0 ± 10.1 µm^2^).

However, no statistical difference was found in the axes of the inscribed ellipsoid in the OL, nor in the OL shape, (the aspect ratios where the osteoporotic OL showed 1% greater a/b, a/c and a/(b + c) and no difference was assessed for b/c).

Moreover, although the mean sphericity index and the mean fractal anisotropy showed no statistical difference between the osteoporotic and the control subject (0% and 1%, respectively), the control showed a much larger variability for both the sphericity index (0.798 ± 0.009 vs. 0.795 ± 0.024) and the fractal anisotropy (0.467 ± 0.015 vs. 0.462 ± 0.023).

The OL volume and surface statistically differed between the osteoporotic subject and the control. Similarly, the second and third principal axes of the inscribed ellipsoid in the OL were different between the osteoporotic subject and the control. However, no statistical difference was identified for region of action, density, porosity, aspect ratios, sphericity index and fractal anisotropy.

In order to assess the potential relationship between the OL characteristics and the bone volume, Spearman correlation coefficients (R^2^) were computed for each VOI. A very strong correlation was found between bone volume and the number of OL (R^2^ = 0.97), while a good correlation was found for the region of action (R^2^ = 0.59). A modest to poor correlation was found for OL volume (R^2^ = 0.36), density (R^2^ = 0.40), surface (R^2^ = 0.27), sphericity (R^2^ = 0.39) and fractal anisotropy (R^2^ = 0.09).

### 3.2. Trabeculae Mechanical Properties

The microindentation assays allowed the characterization of the mechanical properties of the lamellar bone at the micro scale. The results presented in Figure 6 showed that the Es of the control FH had the highest mean value, which was significantly different from the other two regions, 62% higher compared to GT and 47% higher than FN (7.4 ± 2.5 GPa vs. 2.8 ± 1.3 GPa and 3.9 ± 1.2 GPa, *p* < 0.0001, respectively for the Es of FH, GT and FN). Interestingly, the GT showed the lowest Es mean value, which was significantly different (*p* < 0.001) than the Es in the FN area. In the OsteopS group, Es computed in the GT (2.0 ± 0.6 GPa) was significantly different as compared to the corresponding values in FH (3.9 ± 1.3 GPa) and FN (3.5 ± 1.4 GPa). In addition, Es values computed in FH and GT of the control were significantly higher as compared to the corresponding values in the osteoporotic subject (mean Es was 48% higher in FH (*p* < 0.0001) and 29% higher in GT (*p* < 0.01)). Surprisingly, in the FN, Es values were not affected by the osteoporotic status.

As illustrated in Figure 6b, in the control sample, hardness in FH was significantly larger (*p* < 0.0001) than the corresponding values in GT (+32%) and FN (+36%), whereas hardness values in GT and FN were similar. For the osteoporotic sample, hardness differed in FH (0.266 ± 0.099 GPa) and GT (0.215 ± 0.059 GPa). Moreover, similarly to Es intergroup differences, the H intergroup differences presented significant differences in the FH and GT, while no significant differences were obtained in the FN.

The H mean values obtained in the control FH and GT were, respectively, 34% (*p* < 0.0001) and 22% (*p* < 0.05) higher as compared to those from the osteoporotic sample.

### 3.3. ATR-FTIR Results

The ATR-FT-IR spectra of the bones fragment are reported in Figure 7a. One can identify that the main band of phosphates (1010 cm^−1^) of the OsteopS FN and FH were slightly shifted as compared to the controls. The other signals connected to the phosphates (1087, 960, 599, 555 cm^−1^) did not show significant changes. Similarly, the bands of Amide I (1637 cm^−1^) and Amide III (1239 cm^−1^) related to the proteinaceous component remain unchanged.

Some slight differences can be noted in the region 1400–1550 cm^−1^, where the antisymmetric stretching of the carbonate apatite group falls (Figure 7b). The carbonate peak appears in both osteoporotic and control samples and the band at 1407 and 871 cm^−1^, suggesting the presence of B-type carbonated apatite.

All the specimens presented a band at 1444 cm^−1^ followed by a shoulder at 1468 cm^−1^ that could be assigned to the A-type and B-type, respectively [56], although this is with some uncertainty.

The band at 1540 cm^−1^, indicating the presence of A-type carbonated apatite, was not evaluable due to the band overlapping with the Amide II band. The band overlapping represents a well-known difficulty in peak attribution [56], which hampers the recognition of the presence of the A-type apatite. Only the shoulder at 878 cm^−1^ may suggest the presence of the A-type carbonate apatite in all the samples. In summary, the above-mentioned bands could point to the presence of AB-type apatite in all the analyzed samples.

Only in the FN sample of the osteoporotic subject, the amide II band appears different, showing a sharp signal at about 1540 cm^−1^ with additional peaks at 1576 and 1570 cm^−1^. The band’s interpretation at 1540 cm^−1^ seems complicated in this case, as the contributions from the amide II band, the v3 of CAp type-A, and metal carboxylates may occur and overlap. In particular, the presence of metal carboxylates could be supported by the simultaneous signals at 1540, 1576, and 1570 cm^−1^, together with the high intensity of the CH stretching at 2800–3000 cm^−1^. Calcium carboxylates, whose absorption bands are reported in [71] and fall at 1538 and 1576 cm^−1^, could be reasonably formed from the degradation of the fatty tissue of the human body [72].

## 4. Discussion

The multiscale and multimodal investigation of two femoral necks belonging to an osteoporotic subject and a healthy gender- and aged-matched subject undertaken in this study can be of great interest for a reliable assessment of the bone quality using a hierarchical approach to assess bone microarchitecture, trabecular mechanical properties, tissue composition, and OL characterization. The microindentation and the FTIR came to an additional support, respectively assessing the differences in the trabeculae mechanical properties and trabecular Hydroxyapatite composition between different proximal femur regions of the same sample and among different subjects (osteoporotic vs. gender and aged matched control).

This methodology allows the acquisition and the analysis of the femoral neck cortical and trabecular phase, the investigation of OL morphology and organization, the assessment of the local quantitative mechanical properties of the trabeculae and the qualitative characterization of the bone composition and mineralization. This experimental study would provide bone characteristics that cannot be included in clinical studies, although they are fundamental to obtaining a comprehensive overview of the bone quality with direct implications in computational studies [73], pre-clinical research, therapeutic strategy, and eventually clinical practice.

The high -ontrast and spatial resolution SRµCT made the OL visible and characterizable. In this study, OL differences have been assessed between the osteoporotic and the control in the OL volume, surface, and in the total solid phase porosity. However, no differences in the OL shape, sphericity and fractal anisotropy have been observed. Moreover, no statistical difference was assessed, considering the OL as an interconnected network, since OL region of action and density values between the two subjects were in the same range. Previous studies have investigated OL morphology in the cortical phase of femoral diaphysis [13,14,15], proximal tibiae [17] and iliac crest [6], however few studies have assessed the OL morphology in the trabecular bone phase. In a study conducted on the femoral diaphysis of 7 human cadavers (female, aging 75 ± 15 years old) scanned by synchrotron radiation nanotomography (nano-CT) at a voxel size of 120 nm, Peyrin et al. reported a mean OL volume and surface of 315.6 ± 51.7 µm^3^ and 326.0 ± 50.0 µm^2^, respectively [13]. Moreover, the principal axes of the inscribed ellipsoid were a = 16.7 ± 1.8, b = 79.0 ± 1.0 and c = 4.6 ± 0.7 µm. They also reported an OL density of 3.2 ± 1.2 x104 mm^−3^ and average volume of each Voronoi cell or region of action of 2.6 ± 0.6 x104 µm^−3^. The differences in the region of action, which were almost double in our case, could be explained by the different bone region (cortical femoral diaphysis vs. trabecular femoral neck). Similar results were reported by Dong et al. on a study conducted on 13 cortical specimens from the femoral mid-diaphysis of two female donors scanned using SRµCT at a voxel size of 1.4 µm; the mean volume and surface were in the range of 409.5 ± 149.7 μm^3^ and 336.2 ± 94.5 μm^2^, respectively [14]. Moreover, they reported that the average dimensions were of 18.9 ± 4.9 μm in length, 9.2 ± 2.1 μm in width and 4.8 ± 1.1 μm in depth. In another study conducted on proximal tibial trabecular specimens extracted from 3 middle-aged women, respectively affected by osteoarthritis, osteopenia due to rheumatoid arthritis and osteopetrosis, and scanned using an industrial high-resolution nano-CT system at 580 nm^3^, Van Hove et al. reported differences in OL shape between different pathologies, with the osteopenic subject presenting relatively large and round OL [17]. Moreover, they suggested that the differences in 3D morphology of osteocytes and their lacunae in long bones of different pathologies with different BMD might reflect an adaptation to matrix strain due to different external loading conditions. In our case, although the osteoporotic and the control had different BMD, no difference in the OL shape was assessed, which might be due to the fact that both subjects were more than 90 years old and therefore one could suggest a reduced mobility in both cases. To further support this hypothesis, Van Oers et al. reported that osteocytes resulted in alignment to collagen fibres which are affected by the loading mode, moreover suggesting that variation in the lacunar and osteocytes shape undoubtedly affects the osteocytic mechanosensation and subsequent control of bone remodeling [4].

Mechanical loading is an essential stimulus for skeletal tissues. Osteocytes are primarily responsible for sensing mechanical stimuli in bone and for orchestrating subsequent responses. This is critical for maintaining homeostasis, and responding to injury/disease [2]. The microindentation results showed statistical differences in the trabeculae Es and H, depending on both their proximal femur anatomical location and bone health state. The FH showed greater Es and H compared to the other two bone regions investigated, moreover, they also showed a statistical difference between them in terms of bone health state. Although reporting the lowest Es and H, the GT showed similar results to FH. This was not the case for the FN, in which no differences were assessed between healthy and osteoporotic samples.

To the best of our knowledge, trabecular data on osteoporotic bone at the femoral neck could not be found in the literature, and only few studies have been found investigating the trabeculae mechanical properties in this regions [74,75,76]. Hoffler et al. conducted a study on 10 FN trabeculae (male between 40 and 85 years old), in which mean Es of at 8.57 ± 1.2 GPa and H of 0.37 ± 0.05 GPa were assessed. However, differences could be explained by the harvesting of the anatomical site which were closer to the cortical bone than in our case [76]. Moreover, KoKot et al. and Pawlikowski et al., on a study conducted, respectively, on 8 healthy FH (fame and female, 60+ years old) and 25 FH (male and female, 67 ± 9 years old) affected by osteoarthritis, and both using comparable microindentation protocols (trapezoidal loading profile, maximum loading = 500 mN, loading rate = 500 mN/min, holding time = 20 s), reported similar Es and H [77,78]. In particular, the Es and H reported were, respectively, 8.1 ± 1.4 GPa and between 0.30 and 0.50 GPa [77] vs. 7.4 ± 2.5 GPa and 0.40 ± 0.12 GPa in our case for the healthy subject, and 4.8 ± 1.1 GPa and 0.11 ± 0.02 GPa [78] vs. 3.9 ± 1.3 GPa and 0.27 ± 0.05 GPa in our case for bones characterized by an altered remodeling process (osteoarthritis vs. osteoporosis). The slightly lower values obtained in our study could be explained by differences in the preparation protocol (embedded in exothermic resin [74,76] vs. stored in a optimized support in our case) and/or indentation protocol (different maximum loading [75,77,78], holding time of the maximum load [77,78]). Supporting this hypothesis in the review article by Wu et al. has been reported a wide range of bone Es (1.3 GPa and 22.3 GPa) [23], moreover stating that this effect was due to the different scale of analysis (micro, sub-micro and nano scale), different sample preparation (dry vs. wet) [75], anatomical site and localization [79], and orientation of the indentation [80]. The patient age could also be a relevant factor, which could have led to an already altered ECM in both the healthy and the osteoporotic sample via an excessive bone remodeling process due to subjects’ reduced mobility and leading, therefore, to a reduced bone stress profile. The study, conducted by Milovanovic et al. on 8 healthy FN specimens from female donors (5 young donors 32 ± 5 yrs, and 3 elderly donors 88 ± 6 yrs) using nanoindentation, supported this hypothesis and reported that elderly trabeculae expressed less elastic behavior (1.28 ± 0.16 GPa in the young vs. 1.97 ± 0.52 GPa in the elderly) at the material level. This makes elderly populations more vulnerable to unusual impact loads originating from a fall [81]. Therefore, we decided to repeat this microindentation assay on a sample harvested on a younger FN (61 years old), and the mean values obtained for Es were 7.6 ± 0.6 GPa and the H was 0.36 ± 0.05 GPa, which was in the same range found in previous studies [75,76,77]. For these reasons, knowing that mechanical parameters are correlated to ECM, composed by collagen, mineral, and non-collagenous proteins and since no differences have been derived in the FN mechanical properties, one could suggest that the healthy and osteoporotic FN tissues were in the same bone remodeling state. Therefore, one could suggest that healthy FN was already impacted by an intensive bone remodeling process, because FN is one of the first regions impacted by osteoporosis. Overall, the results showed that bone remodeling is a non-uniform process that evolves at different rate depending on the bone anatomical region. An extensive investigation of the OL characteristics in both FH and GT regions could validate this hypothesis and reinforce the link between osteocyte network and bone quality.

The qualitative ATR-FTIR analysis aimed to assess the composition of osteoporotic and control samples, evaluating and reporting their significant differences. In this study, the bands at 1408 and 871 cm^−1^ suggest the presence of B-type apatite, and the band at 878 cm^−1^ suggests the presence of A-type apatite in all the samples. Osteoporotic and control samples show similar spectral behavior, except for sample OsteopS FN, which exhibited peculiar spectral signals in correspondence with the Amide II region. The above-mentioned signals in OsteopS FN might be due to multiple bands overlapping, including also the contribution of carboxylates (bands at 1540, 1570, and 1576 cm^−1^), generated possibly by the interaction of calcium and fatty materials.

Some limitations have to be acknowledged, the first being the use of only two samples and second the synchrotron images focusing only on the FN of the two scanned proximal femurs. The synchrotron beamtime allowed us to fully explore and characterize only one bone region. Therefore, we decided to investigate the proximal femur region more exposed to fractures. A total of 14% of all the fragility fractures occur in the elderly, and are some of the most invalidating, and with high mortality risk (4.3% in-hospital death in patients with mean age of 79.5 years) [18]. Moreover, the two femurs have been accurately selected in order to have very different DXA-BMD, while being extremely alike (gender-, age- and heigh-matched). However, no differences in the OL shape have been assessed in the femoral neck, and as a future perspective we may suggest that it would be interesting to evaluate the OL characteristics also in the great trochanter and the femoral head. In the femoral neck, the differences in the bone mineralized phase suggested by microindentation could be investigated at the molecular scale, as well as for the hydroxyapatite structure, organization and composition.

## 5. Conclusions

In this study, the bone quality in a femoral neck has been characterized using a multiscale and multimodal approach of one osteoporotic subject and one gender- and age-matched control. The proposed method combined X-ray microtomography analyses with Fourier transform infrared spectroscopy and microindentation; this facilitated the investigation of bone quality from the macro scale, by assessing the whole proximal femur region, to the microscale with the assessment of the osteocyte lacunae morphology and organization, and from the trabeculae tissue properties to the molecular scale with the investigation of the mineralization and carbonate accumulation. Our results suggested that small differences could be associated with osteoporosis in the cortical bone phase, while the trabecular network resulted in a greater impact. Moreover, the analysis conducted at the microscale highlighted differences in the trabeculae mechanical properties between the osteoporotic and the healthy subject in both femoral head and great trochanter, while no differences were assessed in the femoral neck in both the mechanical properties and osteocytes lacunae shape and organization. These results suggested that healthy femoral neck was already impacted by an intensive bone remodeling process, and hence femoral neck is the first proximal femur region impacted by osteoporosis. Finally, the qualitative interpretation of FT-IR spectra showed comparable chemical bone composition between subregions. Although a larger study including more samples, both gender and multiple age ranges would be of great interest, this study clearly shows the interest of the bone microarchitecture assessment for evaluating patients’ osteoporotic state.

## Figures and Tables

**Figure 1 materials-15-08048-f001:**
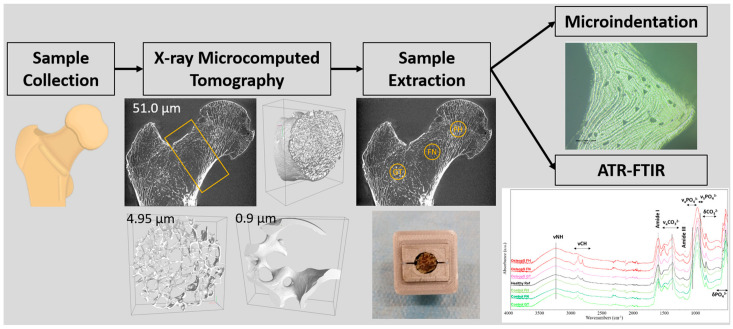
Experimental study workflow.

**Figure 2 materials-15-08048-f002:**
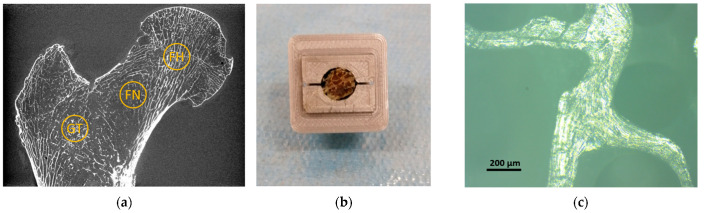
Sample extraction and preparation. (**a**) Localization of the three different bone regions extracted from each proximal femur, “GT” refers to great trochanter, “FN” refers to femoral neck and “FH” refers to femoral head; (**b**) trabecular sample stored in the designed sample holder after the sample preparation protocol; (**c**) optical microscopic investigation used to assess the efficacy of the preparation protocol before the microindentation assays.

**Figure 3 materials-15-08048-f003:**
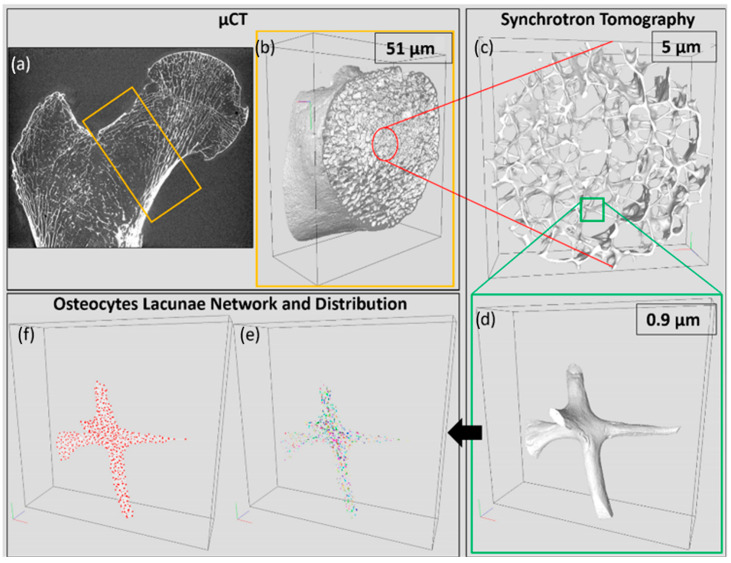
Multiscale characterization of femoral neck morphology acquired using µCT (voxel size: 51.0 µm) and SRµCT (voxel size: 4.95 and 0.9 µm). (**a**) Proximal femur coronal plane acquired using µCT at 51.0 µm showing the femoral neck ROI (orange rectangle) used to assess clinical-standard bone mineral density by dual-energy X-ray absorptiometry; (**b**) whole femoral neck acquired using µCT at 51.0 µm; (**c**) central core of the femoral neck trabecular phase scanned using SRµCT at 4.95 µm; (**d**) single trabecula acquired using SRµCT at 0.9 µm; (**e**) OL distribution and (**f**) OL network.

**Figure 4 materials-15-08048-f004:**
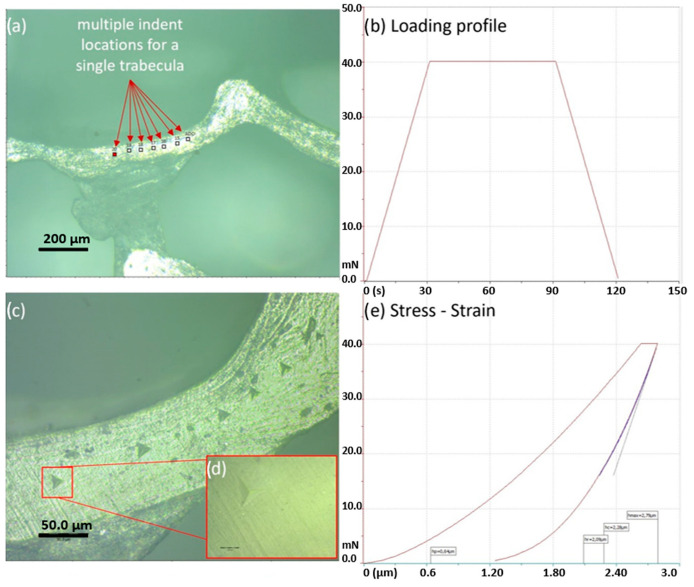
Trabeculae micro indentation. (**a**) Manually placed indent locations for each selected trabecula, (**b**) indentation trapezoidal loading profile, (**c**) stress–strain plot showing a single load–unload cycle, (**d**) indented area and (**e**) view of a single indent at original magnification ×100.

**Figure 5 materials-15-08048-f005:**
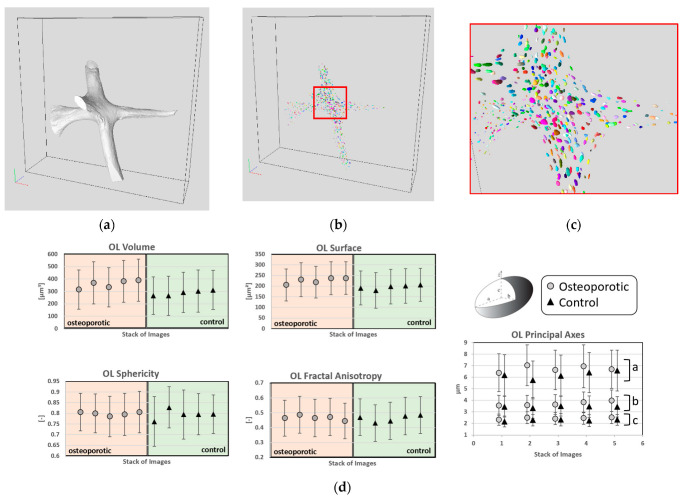
Osteocytes lacunae characteristics. “OL” refers to osteocytes lacunae. (**a**) morphology of a single trabecula, (**b**) visualization of the osteocytes lacunae embedded in the trabecula showed in (**a**), (**c**) magnification of the osteocytes lacunae in the center of (**b**,**d**) morphological analysis showing the volume, the surface, the sphericity, the fractal anisotropy and the principal axes computed from the osteocytes lacunae embedded in the solid bone of both the osteoporotic and the control sample.

**Figure 6 materials-15-08048-f006:**
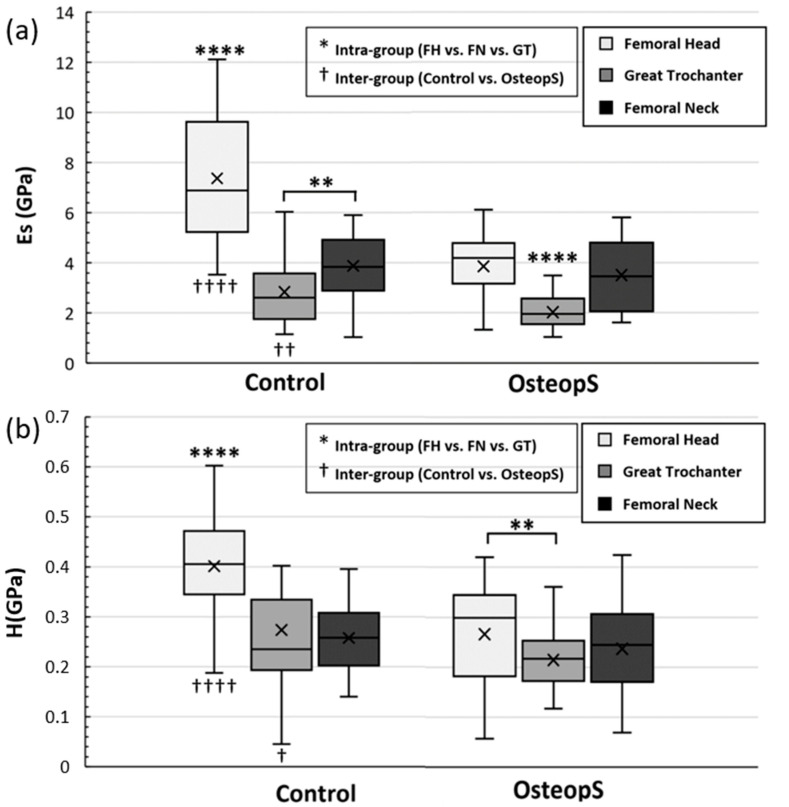
Whisker plot of (**a**) elastic modulus (Es) and (**b**) hardness (H) obtained using microindentation on trabecular bone specimens. (**a**) Representation of intra-group variation of elastic modulus. On the left the differences between the three ROIs in the control subject are presented, and on the right are the differences for the osteoporotic subject. (**b**) Representation of intra-group variation of hardness. On the left are presented the differences between the three ROIs in the control subject and on the right the differences for the osteoporotic subject. * represents the variability between different region of the same group (intra-group), ** *p* < 0.01 and **** *p* < 0.0001. † represents the variability between same regions of different group (inter-group) † *p* < 0.05, †† *p* < 0.01 and †††† *p* < 0.0001.

**Figure 7 materials-15-08048-f007:**
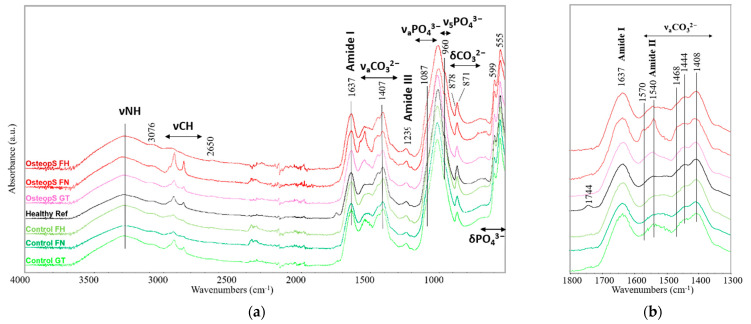
(**a**) ATR-FTIR spectra of the whole set of subjects and subregions. “OsteopS” refers to osteoporotic sample (S08); “Healthy Ref” refers to a refence sample of a healthy 61-year-old male; “FH” refers to femoral head, “FN” refers to femoral neck; “GT” refers to great trochanter. (**b**) Focus in the Amide I, Amide II and carbonate region (1300-1800 cm ^−1^).

**Table 1 materials-15-08048-t001:** Sample description.

	Gender	Age (y)	Leg Pos	Height (m)	DXA (g/cm^2^)		
					Total	Neck	Troch. ^2^
Control	Female	95	Right	1.63	0.939	0.898	0.883
OsteopS ^1^	Female	96	Right	1.65	0.480	0.423	0.419

^1^ OsteopS stands for Osteoporotic sample. ^2^ Troch. Stands for Great Trochanter.

**Table 2 materials-15-08048-t002:** Morphometric parameters of osteocytes lacunae (OL).

**Total Values**		**OsteopS**	**Control**	**Diff**
Nb of Analyzed Regions		5	5	-
Bone Volume (mm^3^)		0.28	0.45	-
OL Number		4030	6649	-
OL Volume (mm^3^)		0.0016	0.0023	-
OL Density (10^4^ mm^−3^)		1.44	1.49	−3%
Bone Porosity (%)		0.59	0.52	+13%
**Mean Values**		**OsteopS (mean ± SD)**	**Control (mean ± SD)**	**Diff**
OL Volume (µm^3^)		358.08 ± 165.00	287.10 **±** 160.00	25% *
OL Surface (µm^2^)		225.53 ± 13.75	195.00 ± 10.11	16% *
Lacunar Density (10^4^ mm^−3^)		1.60± 0.33	1.56 ± 0.16	3%
OL Region of Action (10^4^ µm^−3^)		5.7 ± 2.7	6.0 ± 4.0	−5%
OL Principal Axes (µm)	a (length)	12.13 ± 0.46	11.18 ± 0.57	8%
	b (width)	6.68 ± 0.32	6.19 ± 0.13	8% *
	c (depth)	4.40 ± 0.13	4.09 ± 0.12	8% *
OL Shape (Ad)	a/b	1.90 ± 0.11	1.89 ± 0.09	1%
	b/c	1.54 ± 0.06	1.53 ± 0.06	1%
	a/c	2.85 ± 0.07	2.82 ± 0.17	1%
	a/(b + c)	1.13 ± 0.05	1.12 ± 0.05	1%
OL Sphericity (Ad)		0.80 ± 0.01	0.79 ± 0.02	0%
OL Fractal Anisotropy (Ad)		0.47 ± 0.02	0.46 ±0.02	1%

Data are reported as mean ± standard deviation. “OsteopS” refers to osteoporotic subject, “Diff” stands for difference between the osteoporotic subject and the control, and “Ad” refers to a-dimensional. * indicates a *p* value < 0.05.

## Data Availability

The data presented in this study are available on request from the corresponding author. The data are not publicly available due to restrictions imposed by Aix Marseille University and by the local ethics committee regarding patients data sharing.

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
