# Peer review of "Multiscale Femoral Neck Imaging and Multimodal Trabeculae Quality Characterization in an Osteoporotic Bone Sample"

_materials, 2022, doi:10.3390/ma15228048_

Round 1

Reviewer 1 Report

In the MS, Multiscale femoral neck imaging and multimodal trabeculae quality characterization in a osteoporotic bone sample were performed by a few methods, and obtained some results. It is interesting to some extent. However, several key analyses or investigations should be revised before considering publish. 

In part of 3.3 FTIR spectral analysis, the spectral region of 1400-1600 cm-1 being assigned to carbonate group is incorrect, which should be lower than 1500 cm-1. The authors did not show any related references.

For the same reason, it is not possible to A-type carbonated apatite assigned or overlapped with Amide II.

On the other hand, (line 417), the peak was observed at 1540 cm-1, which was due to “a different molecular order or more probably to the presence of carboxylates as the additional peaks at 1576 and 1570 cm-1 and the high intensity of the CH stretching suggest” (the cited paper is not relative with this MS). This is totally baseless. Actually, there is not so many carboxylates in bone compared to the other components, like collagen, carbonate and hydroxyapatite, etc. It does impossibly dominate the 1540cm-1 peak. Amide II should be analyzed carefully. Additionally, the vibration of CH stretching should be close to 2900cm-1, not around the vibration of amide II. ca. 1550 cm-1 . 

The more important is that the absorbance or IR intensity was analyzed not at all by the author in the whole process, which must be a key factor to investigate the bone disease and component change.

Based on the mentioned above, the reviewer do not think the authors’ conclusion that “The spectral analysis could distinguish neither subregional differences in the osteoporotic sample nor between the osteoporotic and healthy samples” in the Abstract and related analyses are correct. 

The number of sample is just 2, whose statistically significance is not enough for final conclusion.

Reviewer 2 Report

1.      Its existing title's capitalization should be updated to follow the MDPI format.

2.      Emails from all authors are formatted using MDPI and are written in black without italics.

3.      The abstract section should be enhanced to include quantitative data.

4.      As the conclusion of your abstract, please provide a "take-home" message.

5.      Rearrange keywords alphabetically.

6.      Please use lowercase font for each term following MDPI format.

7.      Abbreviation as a keyword is not recommended and encouraged to be changed become a stand for its abbreviation.

8.      Novelty in the current study's is too weak. The past has seen an extensive study of a lot of written material. It is required to provide more details for more explanation about the present novel in the introductory section.

9.      In order to highlight the gaps in the literature that the most recent research aims to fill, it is crucial to review the benefits, novelty, and limitations of earlier studies in the introduction.

10.   Line 91, “In the present study, we aimed…..”. It is not scientific way to mention subject “we”, please make it into passive.

11.   The authors needs to explain the advantage of computational study (in silico) compared with experimental study (in vitro) and clinical study (in vivo). The introduction and/or discussion part of an article should contain this crucial topic, according to the authors. In addition, to reinforce this explanation, the-recommended reference should be cite as follows: Ammarullah, M. I.; Santoso, G.; Sugiharto, S.; Supriyono, T.; Kurdi, O.; Tauviqirrahman, M.; Winarni, T. I.; Jamari, J. Tresca Stress Study of CoCrMo-on-CoCrMo Bearings Based on Body Mass Index Using 2D Computational Model. Jurnal Tribologi 2022, 33, 31–8. https://jurnaltribologi.mytribos.org/v33/JT-33-31-38.pdf

12.   To enhance the understandability of the section on materials and methods easier for them to understand rather than just depending on the main text as it exists at the moment, the authors could add additional illustrations in the form of figures that explain the workflow of the present study.

13.   An evaluation of the findings with similar past research is essential.

14.   Provide a paragraph-length conclusion rather than the present form's point-by-point description.

15.   In the conclusion, please explain the further research.

16.   Five years back literature should be enriched into the reference, and MDPI-published literature is highly recommended.

17.   Due to grammatical and language issues, the authors need to proofread the present work. This problem would use MDPI English editing service.

18.   Ensure that the authors followed the MDPI format exactly, edit the current form, and double-check all of the previously noted problems.

19.   A graphical abstract is suggested to be included in the submission after peer review.

Round 2

Reviewer 1 Report

The revised MS replies all the comments that I concern.  

Reviewer 2 Report

Nice work.